Tour boats affect the activity patterns of bottlenose dolphins (Tursiops truncatus) in Bocas del Toro, Panama

Kassamali-Fox Ayshah akassamali@antioch.edu 1 2
Christiansen Fredrik 3 4
May-Collado Laura J. 5 6 7
Ramos Eric A. 8 9
Kaplin Beth A. 10 11
1 Department of Environmental Studies, Antioch University New England , Keene , NH , USA
2 Panacetacea, St. Paul , Minnesota , USA
3 Aarhus Institute of Advanced Studies , Aarhus C , Denmark
4 Department of Biology, Zoophysiology, Aarhus University , Aarhus C , Denmark
5 Department of Biology, University of Vermont , Burlington , VT , USA
6 Centro de Investigación en Ciencias del Mar y Limnología, Universidad de Costa Rica , San José , Costa Rica
7 Tupper Balboa Ancon, Smithsonian Tropical Research Institute , Panamá , Republic of Panamá
8 Department of Psychology, The Graduate Center, City University of New York , New York , NY , USA
9 Fundación Internacional para la Naturaleza y la Sostenibilidad , Quintana Roo , Mexico
10 Center of Excellence in Biodiversity and Natural Resource Management, University of Rwanda , Kigali , Rwanda
11 School for the Environment, University of Massachusetts at Boston , Boston , MA , USA
Schipper Jan
Electronic publication date: 2020 Mar 30
Publication date: 2020
Volume: 8
Electronic Location ID: e8804
Received 2019 Oct 6; Accepted 2020 Feb 25
Copyright: ©2020 Kassamali-Fox et al.
Copyright year: 2020
Copyright holder: Kassamali-Fox et al.
License: This is an open access article distributed under the terms of the Creative Commons Attribution License, which permits unrestricted use, distribution, reproduction and adaptation in any medium and for any purpose provided that it is properly attributed. For attribution, the original author(s), title, publication source (PeerJ) and either DOI or URL of the article must be cited.
License URL: https://creativecommons.org/licenses/by/4.0/

Keywords: Human interactions, Tourism impacts, Dolphin conservation, Markov chain models, Bottlenose dolphins, Behavioral responses

Funding: Scott Neotropical Fund, Cleveland Metroparks Zoo (2014) The Rufford Foundation (2014) The Waitt Foundation (2014) Panacetacea (2014) Center for Tropical Ecology and Conservation (2013–2014) This work was supported by the Scott Neotropical Fund, Cleveland Metroparks Zoo (2014), the Rufford Small Grants for Nature Conservation from The Rufford Foundation (2014), Rapid Ocean Conservation Small Grant from The Waitt Foundation (2014), Student Research Grant from Panacetacea (2014), and the Tropical Seed Fund Research Grant from the Center for Tropical Ecology and Conservation (2013–2014). The funders had no role in study design, data collection and analysis, decision to publish, or preparation of the manuscript.

==============================
Bottlenose dolphins (Tursiops truncatus) of the Bocas del Toro archipelago are targeted by the largest boat-based cetacean watching operation in Panama. Tourism is concentrated in Dolphin Bay, home to a population of resident dolphins. Previous studies have shown that tour boats elicit short-term changes in dolphin behavior and communication; however, the relationship of these responses to the local population’s biology and ecology is unclear. Studying the effects of tour boats on dolphin activity patterns and behavior can provide information about the biological significance of these responses. Here, we investigated the effects of tour boat activity on bottlenose dolphin activity patterns in Bocas del Toro, Panama over 10 weeks in 2014. Markov chain models were used to assess the effect of tour boats on dolphin behavioral transition probabilities in both control and impact scenarios. Effect of tour boat interactions was quantified by comparing transition probabilities of control and impact chains. Data were also used to construct dolphin activity budgets. Markov chain analysis revealed that in the presence of tour boats, dolphins were less likely to stay socializing and were more likely to begin traveling, and less likely to begin foraging while traveling. Additionally, activity budgets for foraging decreased and traveling increased as an effect of tour boat presence. These behavioral responses are likely to have energetic costs for individuals which may ultimately result in population-level impacts. Boat operator compliance with Panamanian whale watching regulations is urgently needed to minimize potential long-term impacts on this small, genetically distinct population and to ensure the future viability of the local tourism industry.

Introduction

Human-wildlife interactions are on the rise worldwide, generating widespread concern among conservation scientists about the effects of nonlethal human disturbance on the behavioral ecology and conservation status of affected wildlife populations (Duffus & Dearden, 1990; Christiansen et al., 2015). In coastal regions, rising interest in the marine environment has led to an upsurge in boat-based cetacean watching tourism and assumptions about the sustainability of these activities has allowed for tourism operations to proliferate at an accelerated rate (O’Connor et al., 2009). The demand for cetacean watching is especially high in developing countries where regulatory frameworks for managing potential negative effects of tourism are often lacking (Christiansen et al., 2010; Mustika et al., 2013; Pérez-Jorge et al., 2016). Understanding how tourism interactions impact wild cetaceans is vital for protecting threatened populations and ensuring the long-term sustainability of local boat-based tour businesses.

Unregulated tourism activities may impact animals through sublethal energetic effects, challenging the assumption of the sustainability of the industry (Pirotta & Lusseau, 2015; Christiansen et al., 2015; Higham et al., 2015). A number of tourism impact studies have demonstrated an association between cetacean watching and short-term responses in cetaceans including changes in behavior (Lusseau, 2003; Pérez-Jorge et al., 2016), communication (May-Collado & Quiñones Lebrón, 2014; Heiler et al., 2016), respiration and dive characteristics (Stockin et al., 2008; Christiansen, Rasmussen & Lusseau, 2014), swim speed and directionality (Nowacek, Wells & Solow, 2001; Williams, Lusseau & Hammond, 2006; Christiansen, Rasmussen & Lusseau, 2014) and habitat use (Bejder, Dawson & Harraway, 1999; Bejder et al., 2006b; Lusseau, 2005). It is often difficult to correlate short-term behavioral changes with long-term negative outcomes on the biology and ecology of targeted populations, however, there is growing evidence that repetitive short-term behavioral changes, induced by frequent disturbance, can influence life functions (i.e., feeding, predator response, migration and breeding) by imposing additional costs on the energetic budget of individuals (Christiansen, Rasmussen & Lusseau, 2013; Senigaglia et al., 2016; Noren et al., 2016). Tour boat disturbance has been linked to shifts in the activity budgets of numerous cetacean species (Williams, Lusseau & Hammond, 2006; Meissner et al., 2015; Pérez-Jorge et al., 2016; Tyne et al., 2018). These studies indicate that short-term avoidance tactics used by individuals can exert a cumulative effect on animals by altering activity budgets and potentially increasing energetic costs. On the individual level, the physiological constraints produced by lost foraging opportunities and increased traveling activity can lead to long-term negative effects on vital rates and reduced individual fitness. For example, tour boat interactions affected the diving and travelling behavior of female dolphins in Zanzibar, Tanzania (Stensland & Berggren, 2007) and was associated with increased female dive intervals in Fiordland, New Zealand (Lusseau, Slooten & Currey, 2006). A 2016 meta-analysis comparing tour boat disturbance found that disruptions of activity budgets were one of the most consistent responses of cetaceans, further highlighting the use of activity budgets as a metric for evaluation of consequences of tour boat disturbance (Senigaglia et al., 2016).

Bottlenose dolphins (Tursiops spp.) are one of the most targeted small cetacean species for tourism due to their use of inshore habitats and limited movement patterns, making them easily accessible for tourism activities (Samuels et al., 2003). This species is currently listed by the global International Union for the Conservation of Nature (IUCN) as Least Concern (Wells, A Natoli & Braulik, 2019); however, the high geographic and genetic differentiation of the species makes it difficult to assess their actual vulnerability to anthropogenic pressures. In general, coastal populations often face multiple local threats to their conservation, including direct and indirect disturbance and harassment by high levels of boat traffic and commercial dolphin watching activities, and may warrant a higher classification (Hawkins et al., 2017). For example, a quantitative threat assessment concluded that the Fiordland bottlenose dolphin (Tursiops truncatus) subpopulation qualifies as Critically Endangered based on the number of mature individuals and the predicted rate of subpopulation decline over three generations (Currey et al., 2009). Resident coastal dolphin communities are often genetically isolated with restricted home ranges and are at increased risk for the cumulative effects of tourism as individual animals can be subjected to repeated exposure to boat interactions over time (Pirotta et al., 2015). Furthermore, individual animals may be forced to tolerate conditions of high disturbance if they are physiologically or socially constrained to an area or if suitable alternative habitat is lacking (Frid & Dill, 2002).

In Bocas del Toro, Panama, a small, genetically distinct population of coastal bottlenose dolphins is the target of the largest dolphin-watching industry in Panama (e.g., May-Collado et al., 2019; Barragán-Barrera et al., 2017; Sitar et al., 2016. Preliminary analysis suggests the resident population ranges from 72 to 87 dolphins, with 37 individuals exhibiting a high residency rate (38.5 to 100%) in the central dolphin-watching area known as Dolphin Bay ( May-Collado et al., 2019). The dolphins use this embayment for a number of activities including foraging and socializing, and high sighting and recapture rates of females with dependent offspring also suggest it may serve as important nursery habitat for females with calves (May-Collado et al., 2019). Because many of these individuals are resident, they are exposed to tour boats year-round, placing them at increased risk for the cumulative effects of boat disturbance. Previous tourism impact studies at this site have shown that dolphins alter their diving and travelling behavior depending on the number and frequency of tour boat interactions (May-Collado et al., 2015) and modify their acoustic structure (May-Collado & Wartzok, 2008; May-Collado & Quiñones Lebrón, 2014). In 2014, during a series of stakeholder forums held with the local community, at least 12 companies, including cooperatives, tour operators and associations were identified, and a fleet of over 165 boats possessing over 179 captains were estimated to be operating in the Bocas del Toro archipelago and offering dolphin watching tours at that time. It is very likely that the number of companies and boats have increased in the last 5 years. Since 2012, the International Whaling Commission (IWC) has publicly expressed concerns about the potential negative effects of high levels of unregulated boat-based tourism on bottlenose dolphins in Bocas del Toro (International Whaling Commission, 2019). In response to these concerns, the Panamanian government updated its official regulations to manage whale watching activities in the country (Resolución No. DM-0530-2017, Ministry of the Environment, Republic of Panama, 2007). These regulations stipulate that boats should not approach dolphins closer than 100 m and whales closer than 250 m, and that no more than two vessels should interact with one group of whales or dolphins at any one time. The government has also funded training efforts for boat operators to follow best practice guidelines for conducting boat-based dolphin tourism in Dolphin Bay. Despite these efforts, tour boat-dolphin interactions remain high due to lack of boat operator compliance and local government enforcement of tourism activities.

In this study, we assessed the impact of boat-based dolphin watching tourism on the behavioral patterns of bottlenose dolphins in Dolphin Bay, Bocas del Toro, Panama. We aimed to assess if tour boat interactions caused variations in the daytime activity budget of dolphins (hereafter referred to as ‘activity budget’) and whether these changes had the potential to affect dolphins through two vital mechanisms: lost energy acquisition (reduced foraging activity) and increased energy expenditure (increased travelling activity). We hypothesized that significant changes in individual energy gain and expenditure, if frequent enough, could potentially impact survival and reproduction and lead to long-term area avoidance and/or population decline if enough individuals are affected.

Materials & Methods

Study area

The Bocas del Toro Archipelago is located on Panama’s northwestern Caribbean coast near the border of Costa Rica (Fig. 1). The central study area was defined as a 6.3 ×3.1 km area called Bocatorito, also referred to locally as Dolphin Bay, due to the predictable presence of dolphins in the embayment (Fig. 1).

Figure 1 Map of the study site in the Bocas del Toro archipelago, Panama.

This study took place from July to October of 2014 in Dolphin Bay. Dolphin Bay sustains a small, resident population of bottlenose dolphins that is exposed to daily tour boats year-round.

Dolphin Bay is characterized by shallow, clear waters approximately 20 m deep, possessing variable bottom substrate including mud, coral, sea grasses, and mangroves (Kaufmann & Thompson, 2005). Because dolphins use this embayment year-round it is likely an important foraging habitat (May-Collado et al., 2017) where dolphins feed primarily on low-calorie prey such as striped parrotfish (Scarus isiri) and dwarf round herring (Jenkinsia lamprotaenia) (Barragán-Barrera et al., 2019). Since monitoring efforts began in 2004, high sighting rates of mother-calf pairs (including two neonates observed in this study) suggest Dolphin Bay may also serve as critical habitat for females with dependent offspring. A recent study found that resident females possess a relatively long calving cycle (∼62 months) and a high calf mortality rate (46%) (May-Collado et al., 2019). Additionally, nuclear and mitochondrial DNA analyses indicate this population is isolated from others in the Caribbean, including the neighboring population of bottlenose dolphins in Costa Rica located 35 km away from the Bocas del Toro archipelago (Barragán-Barrera et al., 2017). Because the small dolphin community of Dolphin Bay shows such high site fidelity, they have become the main target for dolphin watching activities in the area, resulting in the largest dolphin watching fleet in Panama (May-Collado et al., 2012; May-Collado et al., 2015; May-Collado et al., 2019).

Data collection

The activity of bottlenose dolphins and tour boats was recorded in Dolphin Bay between July and September 2014 (Fig. 1). Surveys were conducted using a small research boat outfitted with a 10 m two stroke outboard engine. The research boat was launched from the dock of the Smithsonian Tropical Research Institute in Saigon Bay at 07:00, 4 to 5 days per week, and stayed out until 1300, weather permitting. We conducted non-systematic surveys of San Cristobal Bay en route to the central study area of Dolphin Bay, and then surveyed Dolphin Bay following a route which included the areas adjacent to the Bay (Fig. 1) until a group of dolphins was found. A group was defined as >1 dolphin moving in a similar direction and engaged in similar behaviors within five body lengths of each other (Mann, 1999).

In the presence of dolphins, the research boat was carefully operated to avoid rapid changes in speed and gear shifts, to minimize potential disturbance to dolphins. The speed of the boat was kept as low as possible to match the speed of the dolphins and to avoid cavitation noise. Additionally, dolphins were always approached from the side and rear. During follows, the boat was maintained in a parallel position to the focal dolphin group at a distance of 20 m or more, both in the presence and absence of tour boats (Nowacek, Wells & Solow, 2001) throughout the data collection process. We could not control for the potential effect of the research boat on the behavior of dolphins, however, dolphin-watching boats in Dolphin Bay routinely approach dolphins at high speeds, at very close range (<5 m) and frequently shift gears, as well as engage in other aggressive tactics. Therefore, it was assumed that the measured effects of the tour boats on the dolphins exceeded the possible effects from the research boat and was hence a conservative measurement.

At the beginning of each dolphin encounter we recorded the time, the location (using a global positioning system), group size, group composition, Beaufort sea state and cloud cover. We attempted to identify all individuals in a group by photo-identification using standard methodology to distinguish individuals with characteristic markings on their dorsal fins (Würsig & Würsig, 1977).

Behavioral sampling

After having recorded the group information and environmental data, individual focal follows were conducted on a single dolphin associated with the focal group. Focal dolphins were selected primarily based on characteristic markings on the dorsal fin. Because the dorsal fin is the most visible and highly recognizable part of the dolphin, dorsal fin identification was used as the primary means of following the focal animal in most observations. However, if prominent scars, rake marks or skin disease were present on the peduncle of the animal and were reliably visible for each sampling period, these markings were also used for following the focal animal, often in combination with dorsal fin identification. Only adults with recognizable marks were selected as focal animals since the behavior of calves was not considered independent of their mothers.

Detailed behavioral information and associated data were collected for the focal animal (e.g., individual description, group size and distance to the nearest tour boat) every three minutes. At the beginning of each sample, one of four activity states was determined and recorded by focal-animal point sampling (Altmann, 1974). Activity states were defined to be mutually exclusive categories of behaviors that as a whole describe the entire activity budget of the focal dolphin. The different activity states for dolphin activity are defined in Table 1 (afterMann & Smuts, 1998; Mann & Sargeant, 2003) and are similar to ones used in other bottlenose dolphin impact assessment studies (e.g., Stensland & Berggren, 2007; Stockin et al., 2008; Christiansen et al., 2010). If the focal animal was not sighted for a maximum of 4 min and the next recorded behavior was the same as that previously recorded, we assumed that the focal animal had been engaged in the same behavior during the missed sampling period. If the behavior had changed between successive recordings the behavior was assumed to have changed during the missed sampling period. Follows were terminated in the event of heavy precipitation and/or lightning, when sea state reached Beaufort 3, visibility deteriorated due to fog or rain, or when the focal animal was lost for >6 min. During surveys, if the same group of dolphins was resighted, we conducted a focal follow on a marked individual we had not yet followed that day. If this was not possible, we searched for another group of dolphins and selected a different individual in that group to conduct a focal follow.

Table 1 Ethogram of activity states for individual dolphins recorded during focal animal sampling.

Activity State	Code	Definition	
Foraging	FOR	Rapid energetic surfacing, frequent directional changes, fish chases, observation of individual with fish in their mouths. Peduncle and tail-out dives frequent (indicating deeper and longer dives).	
Resting	RES	Low level of individual activity, moving slowly (speed <2 kts); slow surfacing 3 to 4 times before diving for an extended period of time.	
Socializing	SOC	Individual engaged in petting, rubbing, mounting, chasing, genital inspections, play, displays, and other physical contact with other individuals.	
Traveling	TRA	Persistent and directional movement (speed >2 kts); individual movement could be meandering but still headed in a general direction.	

Individual follows were classified as either control or impact scenarios. Follows conducted in the presence of the research boat only were treated as control sequences, while follows conducted with the research boat and one or more tour boats were classified as impact sequences. Interactions between tour boats and focal dolphins were defined as beginning when one or more tour boats were within 100 m of the focal dolphin and ended when the last tour boat exceeded this distance. This distance is consistent with the Panamanian government’s whale watching codes of conduct regulations (Ministry of the Environment, Republic of Panama, 2007).

This research was permitted by the Autoridad Nacional del Ambiente, República de Panamá, under permit no. SE/A-55-14, as well as permit no. SE/A 79-14 under Dr. L. J. May-Collado. The research was also approved by the IACUC committee of the Smithsonian Tropical Research Institute, Panama (IACUC-STRI-2011-1125-201406).

Data analysis

Markov chains

We modeled the time-series of dolphin activity states obtained from the point sampling using time-discrete Markov chains (Guttorp & Minin, 1995). We used a first-order Markov chain to assess the difference in transition from the preceding activity state to the immediately succeeding state in the presence (impact) and absence (control) of tour boats. The data obtained from the 3-min focal animal point sampling intervals were first arranged into 2-way contingency tables of preceding activity state versus succeeding activity state (Lusseau, 2003). We developed two contingency tables: one for control and one for impact situations, depending on the presence of tour boats interacting with the focal dolphin between two behavioral samples. If no tour boat interaction occurred between two behavioral samples, the transition between these two samples was placed in a control table (no tour boats present, only research boat). If a tour boat interaction did occur between two samples, the transition was placed in an impact table (one or more tour boats present together with the research boat). When an impact chain followed a control chain, the first transition between them was discarded, as this chain could not be considered as either control or impact since it was not possible to determine the extent of the potential impact (Lusseau, 2003). To test the effect of tour boat presence on dolphin behavioral transitions, the impact and control contingency tables were compared using a Chi-square test. All analyses were completed in R v3.1.2 (R Development Core Team, 2013). The R packages we used for the analyses were “markovchain” (Spedicato, 2017) and “poLCA” (Linzer & Lewis, 2011).

Behavioral transition probabilities

To assess changes in dolphin activity states in the presence of one or more tour boats, transition probabilities from preceding to succeeding activity state were calculated for the control and interaction chains separately, following Lusseau (2003):

pij=aij∑j=1naij,∑j=1npij=1,

where i is the preceding activity state, j is the succeeding activity state, aij is the number of transitions observed from activity state i to j, and pij is the transition probability from i to j in the Markov chain and n is the total number of activity states (i.e., 4). Each transition is the proportion of time a succeeding activity state was observed following a preceding activity state; therefore, we tested the effect of tour boat interactions by comparing the control and impact transition probability matrices for dolphin activity.

Activity budgets

The activity budget for control and impact situations was derived following Lusseau (2003). Differences between the control and impact activity budgets were tested using a Chi-square test, and each activity state in the control activity budget was compared to its corresponding activity state in the impact activity budget using a two-sample test for equality of proportions.

Bout length

The average bout length (the mean duration of time an animal remains in the same activity state) of each activity state, tii was also calculated in the presence and absence of tour boats, following Lusseau (2003):

tii ¯=11−pii,

with a standard error of SE=pii×1−piini

where ni is the number of samples with i as preceding activity. The average bout length for each activity state in both chains was compared using a t-test.

Recovery time

To assess the recovery time, the expected number of transitions it took the dolphins to return to each activity state was approximated for both control and impact chains according to Stockin et al. (2008): ETj=1πj

where (Tj) denotes the time (i.e., number of transitions, i.e., 3 min) it takes a dolphin to return to state j given that the animal is currently in state j and π is the steady-state probability of each activity in the chain. The expected number of transitions was multiplied by the length of each transition unit (3 min) to calculate the average time (min) it took the dolphins to return to each initial activity state. These average times were then compared between control and impact situations to assess the effect of tour boat interactions on dolphin activity states; higher values indicated a longer recovery time for dolphins to return to each initial activity state and lower values indicated a shorter recovery time.

Results

Field effort

During the 10-week study period, we collected a total of 74.9 h of behavioral data from a maximum of 47, and a minimum of 32 different focal dolphins, totaling 79 individual focal follows. Focal dolphins were followed for a minimum of 21 min (7 scans) and a maximum of 60 min (21 scans) under weather conditions which did not exceed Beaufort sea state 3. During this time period, 1,456 behavioral transitions were recorded, of which 829 (57%) were classified as control and 627 (43%) as impact. Control sequences lasted an average of 37 min (median = 36; SE = 3.0; range = 3–96), and impact sequences averaged 36 min (median = 32; SE = 3.6; range = 3–87). The number of boats interacting with focal dolphins at one time varied between 1 and 18, with an average of 2 boats at a time (SE = 0.116). The majority of interactions between tour boats and dolphin groups involved one to two boats (55%). In both control and impact situations, only a low number of transitions between resting and the other activity states were observed (2%), which precluded the use of the resting activity state in the Markov chain analysis. Therefore, any transitions involving resting were omitted and the Markov chain analysis examined only the remaining three activity states (i.e., foraging, socializing and traveling, Table 1).

Effect of tour boat interactions

Behavioral transitions

The results of the Markov chain analyses showed that tour boat interactions significantly affected dolphin activity patterns by altering the transitions between states (Chi-square test, X2 = 34.9251; df = 4; p < 0.001). For all activity states, dolphins were more likely to remain in the same activity state (Fig. 2) than to change to another state (Fig. 2) between time samples. The effect of tour boats was not homogenous over all behavioral transitions. Overall, two behavioral transitions showed statistically significant differences (p <0.05) between impact and control situations (Fig. 3). The probability of going from socializing to traveling (X2 = 4.2156, df = 1, p = 0.0401) increased from 0.176 to 0.378 in the presence of tour boats, whereas the transition probability from traveling to foraging (X2 = 5.0145; df = 1; p = 0.0252) decreased from 0.304 to 0.224 as an effect of tour boat presence (Fig. 3).

Figure 2 Markov chains representing transition probabilities between activity states during control (A) and impact (B) situations.

Values represent transition probabilities and the thickness of the arrows represent the magnitude of these transition probabilities.

Figure 3 Effect of tour boats on the transition probability between activity states for bottlenose dolphins.

Positive values indicate an increase in transition probability when boats were present and negative values indicate a decrease. Significant differences (p < 0.05) are denoted by an asterisk.

Activity budgets

There was a significant difference in dolphin activity budgets between control and impact situations (X2 = 31.0336; df = 2; p < 0.001). Dolphins spent a significantly smaller proportion of their time foraging (X2 = 22.4266; df = 1; p < 0.001) and more time traveling (X2 = 25.5354; df = 1; p ≤ 0.001) in the presence of tour boats (Fig. 4). Foraging was the dominant activity state during both control and impact situations (61% and 48%, respectively), followed by traveling (32 and 45%, respectively). During impact situations, the time budget for foraging decreased by 13% (61% to 48%) and the budget for traveling increased by the same percentage, 13% (32 to 45%).

Figure 4 The activity budget (the proportion of time spent in each activity state) of bottlenose dolphins in Dolphin Bay in the absence (control) and presence (impact) of tour boats.

Error bars represent 95% confidence intervals. Significant differences (p < 0.05) are denoted by an asterisk.

Activity bout length

There was a significant difference in the average length of activity bouts between control and impact situations for all three-activity states (Fig. 5). The length of foraging and socializing bouts were significantly reduced when tour boats were present, whereas traveling bouts were significantly longer. Bout length decreased significantly for foraging dolphins (t = 37.37, df = 729; p < 0.0001) by 3.4 min (95% CI [3.19–3.55]), or 20%, and decreased for socializing dolphins (t = 12.16; df = 103; p < 0.0001) by 3.5 min (95% CI [2.97–4.11]), or 36%. Conversely, the duration of traveling bouts for dolphins increased significantly (t =  − 20.65; df = 461; p < 0.0001) by 2.6 min (95% CI [−2.89–−2.39]), or 30%, during tour boat interactions.

Figure 5 Average bout length (min) for each activity state for control (research boat only) and impact (one or more tour boats present) situations for bottlenose dolphins in Dolphin Bay.

Error bars represent the 95% confidence intervals. Asterisks indicate significant differences (p < 0.05).

Recovery time

The average time it took for dolphins to return to their initial activity state (recovery time) was also altered in the presence of tour boats. Once disrupted by tour boats, foraging dolphins took 1.4 min (29%) longer to return to this state, with an increase from 4.9 min to 6.3 min. Similarly, recovery time for socializing dolphins increased by 3.3 min (8%) when tour boats were present, with an increase from 41.5 min to 44.7 min. In contrast, the time taken to return to traveling decreased with 2.9 min (31%) during tour boat interactions, from 9.5 min to 6.6 min.

Discussion

Effects of tour boats on dolphin activity patterns and activity budgets

The results of this study provide evidence that tour boat interactions significantly affect bottlenose dolphin behavior in Dolphin Bay, Bocas del Toro, Panama. The Markov chain transition analysis showed that the activity states of dolphins were significantly different in the presence of tour boats than during control conditions. More specifically, when interacting with tour boats, dolphins were significantly less likely to begin foraging when traveling and more likely to begin traveling when socializing. Once disrupted by tour boats, dolphins took nearly 30% longer to return to foraging when compared to control conditions, and traveling dolphins more quickly returned to traveling (31%). These changes in activity states were of sufficient magnitude to cause significant changes in the activity budget of dolphins. Overall, dolphins spent a significantly smaller proportion of their time foraging and more time traveling in the presence of tour boats. Furthermore, the average length of time that dolphins spent foraging and socializing was significantly reduced during tour boat interactions and the average time spent traveling was increased, suggesting that dolphin feeding and social behavior was consistently interrupted by tourism and dolphins instead spent more time traveling when tour boats were present, perhaps at the cost of other biologically important activities. Boat-based tourism occurs year-round in Dolphin Bay, where high site fidelity of dolphins likely increases the chance of interactions with specific members of this community. Peak tourism season occurs from December to May during the dry months, while the low-season occurs during the rainy season (June–November). However, up to 39 boats have been reported interacting with dolphins over a period of one-hour in Dolphin Bay during the low season for tourism, demonstrating that dolphins are exposed to high levels of boat tourism year-round (May-Collado et al., 2019). The majority of interactions between tour boats and dolphin groups in this study included 1 to 2 boats (55%), which means that our measured effect could be conservative if we assume that the effect of tour boat interactions increases with the number of tour boats present.

The decrease in time spent foraging during tourism activities could be of biological importance to the Dolphin Bay community. Foraging was the dominant activity for dolphins in this study, therefore, it is reasonable to assume that Dolphin Bay represents an important feeding habitat for dolphins. Barragán-Barrera et al. (2019) found that despite the wide availability of other prey species found throughout the archipelago, the dolphins of Bocas del Toro primarily feed on dwarf round herring (Jenkinsia lamprotaenia), a small and low-calorie prey. Repeated disruption to a critical activity like foraging can result in fewer feeding events, which could lead to reduced energy acquisition for individuals. This could in turn negatively affect the body condition of animals over time, and ultimately impact individual survival and reproductive success (Christiansen, Rasmussen & Lusseau, 2013; Christiansen & Lusseau, 2015; New et al., 2015). The dolphins’ activity budget showed that the proportion of time allocated to foraging under tourism conditions was reduced by the same proportion as their traveling activity was increased, perhaps indicating that dolphins are traveling more to avoid tour boats at the expense of foraging. Studies suggest that reduced energy acquisition resulting from lost foraging opportunities may have a bigger impact on the energy balance and body condition of individuals than energy-costly activities, such as traveling or vertical avoidance (Noren et al., 2016).

The impact of tourism on cetacean social behavior has been given less attention than the effects on other critical behaviors such as foraging, traveling and resting, however, the influence of boat disturbance on dolphin socialization should not be underestimated. Socializing plays an important role in the reproduction of dolphins and disruption to this activity could result in negative impacts to reproductive success (Lusseau, Slooten & Currey, 2006). In this study, socializing dolphins were 20% more likely to switch to traveling in the presence of tour boats and the average length of socializing bouts decreased significantly by 36% in the presence of a tour boat. Previous research in the southern and southeastern United States found bottlenose dolphin social behavior to be highest in the spring and summer months, suggesting that the rate of dolphin socialization may vary seasonally, perhaps as a function of female reproductive receptiveness (Shane, Wells & Würsig, 1986). High levels of boat traffic year-round in Dolphin Bay, therefore, could negatively impact reproduction in this community. More research is needed to identify seasonal trends in dolphin socialization in Dolphin Bay, including identification of a breeding season, as well as the overall social network of the community.

The accepted framework for inferring the costs of tourism on cetacean behavioral ecology suggests that increased energetic challenges, either through reduced foraging opportunities or added traveling costs, can lead to reduced fitness for individuals. If the energetic challenges of avoidance behaviors become too great, or individuals are unable to compensate for these effects, dolphins may shift into long-term avoidance of Dolphin Bay. Based on the high site fidelity of dolphins (population size = 37), Dolphin Bay is a preferred area for dolphins and degradation of this habitat, either through acoustic pollution (e.g., boat engine noise) and/or boat proximity and behavior (e.g., aggressive approaches), could lead to displacement of more sensitive animals from boat disturbance areas, perhaps causing a long-term shift into lower-quality habitat. Dolphin watching tourism has been associated with changes in residency patterns and long-term area avoidance in other geographical areas (Lusseau, 2005; Bejder et al., 2006a). However, it is important to note that inferring the biological relevance of multiple behavioral responses of cetaceans to disturbance is often difficult, especially when considering the high plasticity and behavioral variability of delphinids (Shane, Wells & Würsig, 1986). The Dolphin Bay community may be behaviorally compensating for the observed decline in foraging activity in other ways, for example, by foraging at times of the day not captured by this study or perhaps even at night. Conversely, the resident dolphins of Dolphin Bay do not necessarily reflect a habituated population but instead may reflect a subset of the population that are ecologically or socially constrained to the area and are being disproportionately exposed to tourism activities. Individual responses to disturbance result from a behavioral trade-off process between the costs of staying in a preferred area under conditions of disturbance and the benefits accrued from remaining. Dolphins may perceive tour boats as a form of risk and respond with anti-predator avoidance behaviors, which carry energetic costs, as animals must make behavioral trade-offs between energy acquiring activities (e.g., foraging) and energy-consumptive activities (e.g., traveling) (Gill, Sutherland & Watkinson, 1996). For individual animals with physiologic constraints such as calf dependency, the local abundance of herring as a reliable food source in Dolphin Bay may outweigh the cost of moving to less suitable foraging habitat.

Tourism impact studies of the Dolphin Bay population began in 2006, however, dolphin tourism was present in the archipelago before studies were initiated, therefore it is unknown if the population was always small or if less tolerant animals have already been displaced from Dolphin Bay due to high levels of acoustic and/or behavioral disturbance. In the latter case, tourism activities may already be endangering the viability of this small population because of the increased cumulative exposure that each individual incurs. Individuals unable to flee are more likely to incur the cumulative effects of disturbance and suffer reduced fitness, expressed through decreased reproductive success (Gill, Norris & Sutherland, 2001).

Recommendations for dolphin watching tourism in Bocas del Toro

Currently, the Panamanian government’s whale watching regulations stipulate that no more than two tour boats are permitted to interact with dolphins at one time and boats must maintain a distance of 100 m or more from the animals; however, in a 2013 study of tour boat interactions in Dolphin Bay, three or more tour boats were within distance of 50 m or less 70% of the time, showing a high level of noncompliance with these regulations and minimal action from the Panamanian government in local enforcement, despite repeated recommendations by the IWC (Sitar et al., 2016). In 2012, three calves were killed by boat strikes in Dolphin Bay, and a number of dolphins in this study were observed with new and old propeller-type injuries, suggesting that boat traffic is not only an indirect stressor to dolphins, but also a direct physical threat to their survival (Trejos-Lasso & May-Collado, 2015). Agent-based modeling simulating the effects of cetacean watching on small, genetically isolated populations demonstrate that even low levels of boat disturbance can lead to a decline in population size and possibly even local extinction (Lusseau & Bejder, 2007).

Further assessment of which vital rates are impacted most by disturbance in this population (e.g., survival and/or reproduction) could assist in the development of context-specific tourism management strategies. In New Zealand, an observed decline in local abundance of bottlenose dolphins was correlated with unusual high calf mortality (Tezanos-Pinto et al., 2015). Management strategies that are aimed at protecting the reproductive rate of the Dolphin Bay population could be an effective preventative measure to anticipated decline of this population. For example, eliminating or strongly limiting tour boat interactions with groups containing mother-calf pairs should be strongly considered. Additionally, delineation of protected zones for dolphins based on their habitat preferences for critical biological activities (e.g., foraging, socializing and resting) could be of importance in minimizing disturbance to foraging activities of dolphins in Dolphin Bay, as well as affording ample space for undisturbed socialization activities.

Ecotourism is a highly profitable industry in Bocas del Toro, with the potential to provide economic benefits to local communities as well as promote conservation of the marine environment. However, the rapid, uncontrolled growth of tour boat activities and the aggressive practices of some tour boat operators around dolphin groups is currently of high conservation concern. In 2015 the IWC called for the Panamanian government to enforce its whale watching regulations, and there has since been a significant effort to educate and train boat operators on how to safely operate boats around dolphins. However, three major issues remain unaddressed: (1) the high turn-over of boat-captains dilutes training efforts, (2) the current lack of a licensing process promotes high boat traffic and low rates of compliance, and (3) boat tours do not necessarily reflect tourists interests (e.g., tourists prefer to see sloths and monkeys than dolphins) (May-Collado et al., 2015; May-Collado et al., 2019). The failure to address these issues (despite increased boat operator training efforts) means that dolphins continue to be affected by high levels of boat tourism, highlighting the need for urgent reform of tourism practices and marketing to protect the viability of the local dolphin population and the tourism industry (Sitar et al., 2016).

Boat operators are not currently licensed in Bocas del Toro; therefore, establishing a permitting process for dolphin watching tourism appears to be a reasonable next step. However, more governmental involvement is needed for effective enforcement of permits. Currently, collaboration between scientists, governmental officials, community leaders and local stakeholders are paving the way for a community-based tourism management initiative through development of a “bottom-up” boat operator training and permitting program that will help limit the number of tour boats in Dolphin Bay, as well as empower licensed operators with the skills and training needed to conduct tourism experiences based on best practices. Effectiveness of these strategies should continue to be monitored and supported as well as continuing to appeal to the Panamanian government for support and oversight. Further consideration should also be given by the “international” IUCN to evaluate the conservation status of this small, genetically distinct community of T. truncatus to reflect its actual threat level based on the behavioral results of this study, as well as a number of other studies demonstrating low genetic diversity of this community and tour boat effects on short-term behavior and acoustic structure.

Conclusions

The results of this study demonstrate that current levels of dolphin watching tourism in Dolphin Bay are affecting dolphin activity patterns in sufficient quantity to alter their activity budgets. Dolphins foraged less and traveled more in the presence of tour boats, the average length of time that dolphins spent foraging and socializing was significantly reduced, and the average time spent traveling increased during tour boat interactions. These changes in activity patterns may ultimately carry energetic costs for individuals and have long-term negative implications for individual vital rates and population dynamics. The energetic consequences of reduced foraging and increased traveling activity are likely to be more pronounced for mother-calf pairs who are already constrained by their reproductive physiology. More research into maternal behaviors and calf survivability is needed to understand the relationship between energetic costs and boat tourism effects. The behavioral results from this study provide further evidence that the Bocas del Toro bottlenose dolphin population is highly vulnerable to the cumulative effects of tourism activities, highlighting the need for identification of “at-risk” conservation management units for this small, genetically distinct population.

Supplemental Information

File S1 The database of dolphin sightings and behavior analyzed in this study

Click here for additional data file.

File S2 The R code for the Markov Chain analysis

Click here for additional data file.

Supplemental Information 1 Frequency of the number of tour boats per encounter with dolphins

Click here for additional data file.

A big thank you to the STRI Bocas del Toro field station and staff for providing a research boat, logistical support and a great lab space to work in. Thanks to the boat captains who assisted with this project, thank you for your knowledge and hard work. A special thank you to our boat captain D. Georget, for his patience with the research process and for his dedication to the conservation of the Bocas dolphins. Thanks to all of the inspirational ladies of Panacetacea’s Bocas Dolphin research team: S. Quinones-Lebron, D. Barragán-Barrera, M. Gamboa-Poveda and L. Trejos. A big thank you to our field assistants Cole, Giselle and Katherine (KC) for all of their hard work and dedication to data collection, and to advisors, colleagues and friends for their invaluable support in data analysis and reviewing this manuscript.

Additional Information and Declarations

Competing Interests

Author Contributions

Animal Ethics

Field Study Permissions

Data Deposition

The authors declare there are no competing interests.

Ayshah Kassamali-Fox conceived and designed the experiments, performed the experiments, analyzed the data, prepared figures and/or tables, authored or reviewed drafts of the paper, and approved the final draft.

Fredrik Christiansen, Laura J. May-Collado and Eric A. Ramos analyzed the data, prepared figures and/or tables, authored or reviewed drafts of the paper, and approved the final draft.

Beth A. Kaplin conceived and designed the experiments, authored or reviewed drafts of the paper, and approved the final draft.

The following information was supplied relating to ethical approvals (i.e., approving body and any reference numbers):

This study was approved by the IACUC committee of the Smithsonian Tropical Research Institute, Panama (IACUC-STRI-2011-1125-201406).

The following information was supplied relating to field study approvals (i.e., approving body and any reference numbers):

This research was permitted by the Autoridad Nacional del Ambiente, República de Panamá, under permit no. SE/A-55-14, as well as permit no. SE/A 79-14 under Dr. L. J. May-Collado.

The following information was supplied regarding data availability:

The database of dolphin sightings and behavior and the R code for the Markov Chain analysis are available in the Supplemental Files.

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
