# Peer review of "Tour boats affect the activity patterns of bottlenose dolphins (Tursiops truncatus) in Bocas del Toro, Panama"

_PeerJ, doi:10.7717/peerj.8804_

## Round 0.1 · original submission · Minor Revisions

Thank you for your submission. As you will see the reviews are very favorable but make some important suggestions which I feel will improve the final product. We look forward to seeing an updated version considering these minor revisions.

Reviewer 1 ·

Basic reporting

This manuscript is very well written - clear, concise, and professional. The topic is important and timely and will make a nice contribution.

There are a couple of areas that need support from references (included in the attached pdf). There are also a couple of issues with citation formatting.

I suggest replacing the use of "impact" with "effect", in several instances (see attached pdf), which are often misused in animal behavior literature. Effect may be more appropriate for the short-term behavioral patterns that are described in this manuscript. For clarification, please refer to: Beale. 2007. The Behavioral Ecology of Disturbance Responses. International Journal of Comparative Psychology 20.

Another terminology issue exists - code of conduct "recommendations" are different than laws/regulations. Recommendations are not enforceable because they are not required by law. Please clarify in the inrtoduction and discussion sections what is currently in place - if recommendations only, then your text about compliance and enforcement in the discussion section do not make sense in this context.

Experimental design

This original research has well-defined questions and methods that are mostly described with sufficient detail. However, there is a huge range of tour boats that were considered "present" with dolphin groups, ranging from 1-18. Please include more explanation in the methods and results - for example, what is the sample size breakdown of tour boats per group - how many sequences with 2, 3, 4, etc. tour boats? Binning all of the boat number categories together as "present" or "absent" may skew results. There are examples in the literature that show that the NUMBER of tour boats may affect dolphins differently (e.g., Bejder, L., A. Samuels, et al. (2006). "Decline in relative abundance of bottlenose dolphins exposed to long-term disturbance." Conservation Biology 20(6): 1791-1798.)

Word choice when describing Chi-square results should be addressed. Chi square results are presented as one thing occurring more or less than "expected by chance". It is not a direct measure of one thing occurring more or less than another thing. See McHugh, M. L. (2013). The chi-square test of independence. Biochemia medica: Biochemia medica, 23(2), 143-149.

Validity of the findings

See comments in section 2 (Experimental design) on number of tour boats and how they are handled in the analyses.

Annotated reviews are not available for download in order to protect the identity of reviewers who chose to remain anonymous.

Reviewer 2 ·

Basic reporting

The manuscript is logically and clearly written and provides sufficient background information to be informative.
The study covers a clear and important question.

The English language is good and ensures that an international audience can clearly understand the text. There are some minor parts where the English language should be improved to increase the article's clarity. One examples where the language could be improved includes lines 159 (“a 10m two stroke engine”) and 378, where the current phrasing makes comprehension difficult.
A detailed background supported with strong references is provided.

The raw data is present, as well as the R script of the analysis. Both are well organised.
The hypothesis is clearly outlined, tested, and discussed using the data collected.

Experimental design

The experimental design is described in detail with a clear structure.
The research question is relevant and meaningful.
There are two areas where I would appreciate further detail. It was unclear to me what was meant by “A group was defined as > 1 dolphin” on line 165, as in line 169 it is written “the speed of the dolphin(s)”. Does it mean that you did not take into account if there was a single dolphin, which would make sense as a single dolphin would not show a behaviour change towards socializing? Also, what did you do in case you had a resighting of the same group of dolphins? Please add a more detailed description.

The statistical methods used are fully described in the article. The R packages used in the statistical analysis are not presented or cited in the article. R packages within R can be cited using the code "citation(package = "PACKAGENAME")" for example "citation(package = "poLCA")".
In line 243 you should use correct referencing, instead of “Lusseau (2003)” use (Lusseau, 2003) or “…following Lusseau (2003):”

Validity of the findings

The findings are solid, and are an important investigation of an increasing problem in the field of marine mammal conservation. I think the results are communicated with clear figures and tables, helping to understand paragraphs of text with reporting of parameters.
All underlying data have been provided, they are robust, statistically sound, and controlled.
Results are accurately presented and conclusions are linked to the original research questions.

---

## Round 0.2 · accepted · Accept

We appreciate your diligence in addressing all of the suggestions and I think the current version reflects that. Thank you for submitting to PeerJ.